# Fish Scale-Inspired Flow Control for Corner Vortex Suppression in Compressor Cascades

**DOI:** 10.3390/biomimetics10070473

**Published:** 2025-07-18

**Authors:** Jin-Long Shen, Ho-Chun Yang, Szu-I Yeh

**Affiliations:** Department of Aeronautics and Astronautics, National Cheng Kung University, Tainan City 701, Taiwan; wklaelul@gmail.com (J.-L.S.);

**Keywords:** fish scale structure, compressor cascade, corner separation, passive flow control, total pressure loss

## Abstract

Corner separation at the junction of blade surfaces and end walls remains a significant challenge in compressor cascade performance. This study proposes a passive flow control strategy inspired by the geometric arrangement of biological fish scales to address this issue. A fish scale-like surface structure was applied to the suction side of a cascade blade to reduce viscous drag and modulate secondary flow behavior. Wind tunnel experiments and numerical simulations were conducted to evaluate its aerodynamic effects. The results show that the fish scale-inspired configuration induced climbing vortices that energized low-momentum fluid near the end wall, effectively suppressing both passage and corner vortices. This led to a reduction in spanwise flow penetration and a decrease in total pressure loss of up to 5.69%. The enhanced control of secondary flows also contributed to improved flow uniformity in the end-wall region. These findings highlight the potential of biologically inspired surface designs for corner vortex suppression and aerodynamic efficiency improvement in turbomachinery systems.

## 1. Introduction

Corner separation is a typical flow phenomenon occurring at the junction of blade suction surfaces and end-walls in axial compressors [1]. This phenomenon significantly deteriorates compressor performance by inducing passage blockage, reducing static pressure rise, increasing total pressure loss, and lowering overall efficiency. The achievement of high-pressure ratios, a critical requirement for high-performance compressor stages, necessitates the effective control of internal flow separation during the design process. Effective flow control can mitigate corner separation, reduce aerodynamic losses, and enhance the overall efficiency of axial compressors.

Flow control techniques are implemented in the corner regions of the cascade and are generally divided into active and passive categories. Active flow control technologies include methods like plasma actuation [2], end-wall suction [3], pulse jets [4], and synthetic jets [5]. These methods involve controlling the flow field by adding energy through auxiliary equipment, such as fluid injection or external force application. Nonetheless, this also increases the complexity of the design and manufacturing process. Alternatively, passive flow control technologies encompass non-axisymmetric end-walls [6], vortex generators [7], blade tip winglets [8], end-wall fences [9], and blade ribs [10].

Natural organisms possess specific flow control mechanisms for them to adapt to changes in their environment. These mechanisms have served as inspiration for the construction of passive flow control devices. Wang et al. [11] examined aerodynamic performance cascade configurations featuring smooth and non-smooth surfaces, inspired by the microgroove structures found in shark skin, under varying pressure gradients. Their simulation indicated that the non-smooth structure effectively reduced skin friction and turbulence intensity within the flow channel, leading to a significant reduction in total pressure loss. Dong et al. [12] proposed a novel technique profiling leading-edge tubercles based on attenuation functions and sine waveforms inspired by the morphology of Humpback whale flippers. Their approach allowed for the adjustment of tubercle amplitude and wavelength through a sine function, revealing that smaller amplitudes could minimize additional losses in the mid-span region, while larger wavelengths promoted separation vortex formation. Lu et al. [13] explored the impact of ellipsoidal dimples arranged in columns on cascade performance. Their findings showed that the inclusion of these dimples enhanced energy exchange between the mainstream flow and the low-energy region within the boundary layer near the wall, improving overall aerodynamic performance. This improvement enhanced the fluid’s anti-separation capability. It reduced total pressure loss by suppressing the migration of spanwise secondary and transverse flows, thereby decreasing the range and intensity of the corner and passage vortices. In their study, the authors designed bio-inspired structures of the cascade to reduce losses, but they discovered that the microgroove structure and the surface of the tubercles were excessively complex and expensive to machine.

Benner et al. [14,15] proposed an empirical approach to predict secondary losses in turbine cascades. Their analysis highlighted the influence of spanwise penetration depth, defined as the spanwise distance between the separation line and the end-wall at the trailing edge, on total pressure loss. Total losses were attributed to secondary losses, profile losses, and tip-leakage losses, with a strong positive correlation identified between profile losses and total losses. A reduction in the spanwise penetration depth of the climb secondary flow was found to decrease total pressure loss, thereby enhancing cascade performance. Shan et al. [16] investigated the application of a micro-vortex generator upstream of the corner vortex (CV), particularly at a larger incidence angle, which effectively delayed stall onset. For the baseline cascade, this delay was noted at an incidence angle of 7.9°. Moreover, the addition of a vortex generator not only eliminated the stall behavior but also significantly reduced the total pressure loss. Cao et al. [17] examined the control mechanism of an end-wall passage vortex generator (EVG) on the corner stall of a high-load cascade. They utilized vortex visualization to determine the position of the vortex structure. They used three-dimensional streamlines to observe the trajectory of the vortex and understand how the end-wall passage vortex generator improved the flow field in the cascade passage. In their study, they placed the end-wall passage vortex generator upstream of the corner vortex. This technique effectively mitigated the secondary flow losses in the cascade’s flow field.

Overall, we examined the literature on the effects of vortex structures on the distribution of total pressure loss. Many studies employ advanced vortex visualization techniques combined with projected 2D streamline analyses to thoroughly investigate the formation, development, and evolution of vortical structures within the flow field. These approaches enable researchers to clearly reveal complex 3D vortex structures and to better understand how such structures influence fluid dynamic characteristics and the spatial distribution of energy losses. This multifaceted analysis provides an important theoretical foundation and valuable references for the present study.

In our previous study [18], we developed a single array of grass carp fish scales positioned upstream of the corner vortex to control corner separation and improve compressor cascade performance. This design aimed to reduce geometric complexity and computational demands while effectively addressing flow losses. This study builds on previous work by employing a fish scale array with low viscous friction drag to further control corner separation and improve compressor cascade performance. Using a combination of numerical simulations and wind tunnel experiments, we analyzed secondary flow structures through techniques such as vortex visualization, helicity analysis, axial velocity density (AVD), projected 2D streamlines, and spanwise penetration depth measurements. The cascade flow field was divided into four regions—the end-wall, passage, blade profile, and wake—to quantify total pressure loss and optimize the cascade configuration. These findings offer insights into applying additional bio-inspired features to mitigate adverse effects like stall, passage blockage, and flow separation, thereby enhancing compressor efficiency and performance.

## 2. Methodology

### 2.1. Experimental Setup

Experiments were carried out in a low-speed linear compressor cascade wind tunnel at National Cheng Kung University in Taiwan, as illustrated in Figure 1. In the test section, the free-stream turbulence intensity was measured to be below 0.7% within a velocity range of 5–20 m/s. Tailboards were installed at the trailing edges of the first and seventh blades to ensure uniform inflow and maintain periodic flow conditions in the cascade.

A five-hole probe with a cone angle of 45° and a tip diameter of 3 mm was used to determine the static pressure, total pressure, and velocity in the cascade’s flow field. The probe was attached to a three-axis linear guide system to facilitate area traversal and analyze the 3D flow structures within the cascade. Traversal was conducted in various planes downstream of the blade’s trailing edge. An AMS-5812 pressure sensor (Analog Microelectronics, Mainz, Germany) was employed as a pressure transducer in the voltage range 0.5–4.5 V with an uncertainty of ±1.5%. This sensor measures pressure within ±1000 Pa by transforming pressure into voltage signals. Oil-film visualization was used to analyze the flow patterns on the blade’s suction surface. A thin layer of a mixture consisting of paraffin oil, oleic acid, titanium dioxide powder, and silicone oil was applied to the surfaces. Following approximately 30 min of operation in the cascade wind tunnel, the oil film was carefully removed, and the resulting patterns were documented through photography.

#### 2.1.1. Cascade Description

The cascade blades used in this study feature a widely utilized NACA65810 airfoil, commonly referenced in the literature [19]. The schematic of the blade geometry is shown in Figure 2, along with the designed inlet velocity of 20 m/s, the specific geometric parameters, and the relative operating conditions employed throughout both the experiments and numerical simulations in this study.

In our previous study, various fish scale structures were tested on a flat plate under the same inlet flow condition of 20 m/s to evaluate their geometric performance [18]. The parameters of these structures were systematically varied to identify the configuration with the lowest viscous drag reduction rate (*DV*). The optimal fish scale structure, with a height *h* = 0.4 mm and an adjacent spacing *L_s_* = 2 mm, is illustrated in Figure 3a. This configuration was found to generate climbing vortices, which enhanced the kinetic energy of the fluid near the wall and reduced *DV* by 21.96%. Some studies [20,21,22] indicated that a vortex generator should be placed in front of the corner vortex to suppress corner separation. Building on this idea, we developed a fish scale structure array to replace the vortex generator. In addition to the original cascade (ORI case), we designed three fish scale structures to be placed on the blade’s suction surface. We placed the fish scale array with one row and seven columns across 15% to 30% (Case 1), 35% to 50% (Case 2), and 65% to 80% (Case 3) of the chord on the suction surface (Figure 3).

#### 2.1.2. Periodicity Validation

To confirm the periodicity of the flow field, a five-hole probe was used to measure the total pressure loss distribution at the cascade outlet. The probe was placed at the middle span of the blade, 0.1*c* away from the trailing edge of the blade. This verification ensured the feasibility of applying periodic boundary conditions to reduce computational effort without compromising accuracy. Figure 4 illustrates the distribution of the total pressure loss at the mid-span of the outlet for the ORI case, under conditions of *V* = 20 m/s and *α* = 15° along the grid pitch. A comparison of the wake profiles on either side with the mainstream parameters confirmed that the tailboard-equipped cascade exhibited highly periodic outlet airflow.

### 2.2. Numerical Methods

This study involved both numerical simulations and experiments. ANSYS 19.2 Fluent software [23] was employed to simulate the flow over a flat plate and evaluate the flow losses in the compressor cascade. The computational domain covered half of the experimental flow domain in the spanwise direction, with symmetry boundary conditions applied at the mid-span [18]. DeGroot et al. [24] analyzed turbulence modeling in flow channels and noted that Reynolds Averaged Navier–Stokes (RANS) models offer significantly lower computational cost compared to Direct Numerical Simulation (DNS). Therefore, the cascade numerical in this study was carried out using the *k-ω* SST (Shear Stress Transport) turbulence model.

Figure 5 illustrates the numerical and experimental oil-film visualization of limiting streamlines on the suction surface of the blade. The results indicate a strong correlation between the numerically determined position of the corner region and the oil-film distribution, confirming the accuracy of our calculation approach in predicting the separation line position on the suction surface. These findings validate the reliability of our numerical simulation. Figure 6 presents a comparison of the total pressure loss distribution at *x*/*c* = 1.1 downstream of the cascade outflows, with experimentally measured coefficients on the left and the numerically estimated coefficients on the right. In the tail region of the blade, the numerically estimated total pressure loss closely matched the experimental results. The RANS Shear Stress Transport *k-ω* (RANS SST *k-ω*) turbulence model accurately captured the location of the separation line and significantly reduced the computational time for simulations. Many studies [25,26] have used RANS, LES, and DNS for flow field simulations. Although some studies [27,28] have demonstrated that LES and DNS can accurately capture flow field and turbulence characteristics, these methods incur high computational costs and require extensive computational time. Therefore, we selected the RANS SST *k-ω* turbulence model for our bio-inspired cascade investigation.

#### 2.2.1. Boundary Condition

The inlet velocity and turbulence intensity, as determined experimentally, were prescribed at the boundary, while the outlet was set as the atmosphere boundary. The blade’s surface and end-wall’s surface were modeled with no-slip boundary conditions, while both the right and left walls utilized translational periodic boundary conditions. The computational grid near the end-wall and the fish scale structure was refined to accurately capture complex secondary flow structures in the cascade, ensuring that the y^+^ value was close to 1, as required for the Shear Stress Transport *k-ω* turbulence model (Figure 7).

#### 2.2.2. Grid Independence Test

The numerical simulations were conducted across the entire computational domain using different grid resolutions to confirm that the results were independent of the computational grid density. The total pressure loss coefficient (*C_pt_*) represents the flow losses in the compressor cascade and is expressed as(1)$$C_{pt}=\frac{P_{t,in}-P_{t,local}}{P_{t,in}-P_{s,in}}$$
where *P_t_*_,*in*_ and *P_s_*_,*in*_ denote the total pressure and static pressure of the inlet flow, respectively. *P_t_*_,*local*_ refers the local total pressure.

The effect of varying grid resolutions (i.e., number of cells) on the total pressure loss coefficient distribution at *x*/*c* = 1.1 downstream of the cascade is shown in Figure 8. The results include both the baseline case without fish scale structures (ORI case) and case with fish scale structures (Case 3). When the cell number was increased from 4.5 million, the total pressure loss coefficient at the cascade outlet showed minimal variation. Hence, to ensure computational accuracy while optimizing resource use, a grid resolution of 4.5 million cells was selected for subsequent numerical calculations.

#### 2.2.3. Validation of Numerical Simulations

The validation presented in Section 2.2.2 specifically refers to the baseline case (i.e., ORI case) without the fish scale structures which ensured the accuracy of the numerical model before applying it to the biomimetic configurations. Figure 9 illustrates the total pressure loss coefficient distributions along the spanwise direction, as determined from numerical simulations and experiments. Overall, the numerical simulation results show good agreement with the experimental data for both the transient state and steady state. Notably, for the simulation, a time instant at *t* = 0.07 s was selected, as the flow field reached a quasi-steady state at *t* = 0.06 s. After this point, the transient results showed negligible variation and were found to be nearly identical to the steady-state results. The transient simulation also helped to understand the formation and evolution of vortex structures. Therefore, we chose the transient results as the computational model for this study.

### 2.3. Metrics for Vortex Visualization

Helicity quantifies the corkscrew-like motion of a fluid, occurring when a fluid parcel undergoes a solid body rotation around an axis aligned with its direction of motion. In fluid dynamics governed by the Euler equation, both vorticity and helicity are conserved quantities and can be used to analyze end-wall flow structures. Helicity is mathematically defined as the dot product of velocity and vorticity, expressed as(2)$$H=\left(\nabla \times \vec{V}\right)\cdot \vec{V}$$
where $\left(\nabla \times \vec{V}\right)$ represents the vorticity, and $\vec{V}$ denotes the velocity vector.

Fish scale-inspired structures have a significant impact on flow behavior, altering both flow losses and the capacity for diffusion. Axial velocity density (AVD) is used to identify the volume of low-energy fluid enclosed by an iso-surface. AVD is defined as(3)$$\mathrm{AVD}=\rho \times V_{a}$$
where *ρ* represents the density and *V_a_* denotes the axial velocity.

The *Q*-criterion is used to represent the vortex structures within the cascade passage [29]. The following is the definition of the *Q*-criterion:(4)$$Q=\frac{1}{2}\left({\left\|\Omega \right\|}^{2}-{\left\|S\right\|}^{2}\right)$$(5)$$\frac{\partial u_{i}}{\partial x_{j}}=S+\Omega$$(6)$$S=\frac{1}{2}\left(\frac{\partial u_{i}}{\partial x_{j}}+\frac{\partial u_{j}}{\partial x_{i}}\right)$$(7)$$\Omega =\frac{1}{2}\left(\frac{\partial u_{i}}{\partial x_{j}}-\frac{\partial u_{j}}{\partial x_{i}}\right)$$
where Ω denotes the vorticity tensor, *S* represents the strain rate tensor, and *Q* refers to the velocity gradient tensor.

Vortex identification in this study primarily relies on the *Q*-criterion technique, which distinguishes vortex cores by evaluating whether local rotation dominates over strain, i.e., regions where *Q* > 0. In our visualizations, vortex structures are represented by iso-surfaces of *Q*, while the coloring of these surfaces is based on helicity values to indicate the rotational direction of each vortex. This combination allows us to distinguish different types of vortices, such as passage vortices and corner vortices, based on their geometry and helicity sign.

To further analyze the aerodynamic implications of these vortex structures, we utilize axial velocity density (AVD) to identify low-energy-fluid regions and the total pressure loss coefficient to evaluate overall flow losses. The integrated use of *Q*-criterion, helicity, AVD, and total pressure loss provides a comprehensive view of how vortex dynamics affect the energy distribution and aerodynamic performance within the cascade. This multi-faceted approach is essential for understanding the interactions between secondary flows and loss mechanisms.

## 3. Results and Discussion

To determine the influence of secondary flow losses and total pressure loss, we use helicity (with positive values represented counterclockwise and negative values represented clockwise), as well as projected 2D streamlines, to visualize the direction of rotation of the vortex. These metrics are combined to assess the contribution of vortex structures to flow losses. Figure 10 illustrates the total pressure loss distribution at *x*/*c* = 1.1 downstream of the cascade outflow, along with helicity and the secondary flow distribution. In Case 1 and Case 2, the high total pressure loss region exhibited minor changes, indicating that the fish scale structure did not effectively suppress the passage vortex, leaving the total pressure loss distribution essentially unchanged. In contrast, Case 3 showed a significant reduction in the high-total-pressure-loss region. A detailed comparison of the helicity and projected 2D streamline distributions in Case 3 revealed that the passage vortex (PV) and concentrated shedding vortex (CSV) were influenced by the fish scale structure, which in turn caused the vortex structure to decrease and move toward the end-wall.

Figure 11 compares the spanwise variation in the pitch-averaged total pressure loss coefficient for the four cascade configurations. The results indicate that placing the fish scales near the leading edge of the suction surface (Case 1) failed to suppress the passage vortex, increasing total pressure loss. In Case 2, a mild suppressive effect on the passage vortex was observed, with a slight decrease in the total pressure loss coefficient in the spanwise direction from *z*/*s* = 0 to *z*/*s* = 0.1. In Case 3, the passage vortex structure was significantly suppressed, resulting in a notable decrease in the total pressure loss coefficient over the corresponding spanwise range.

Various vortex structures were identified in the flow field, namely the passage vortex (PV), corner vortex (CV), wall vortex (WV), concentrated shedding vortex (CSV), and trailing edge vortex (TEV). To investigate the impact of the fish scale array on cascade performance, the underlying mechanisms of these vortices were examined. Figure 12 shows the vortex structure near the corner region in the cascade, visualized using *Q* = 2.5 × 10^4^ s^−2^ and color-coded based on helicity. The horseshoe vortex (HV) was formed due to the presence of an adverse pressure gradient, causing the boundary layer fluid to roll up at the leading edge. The HV divided into two branches in front of the stagnation line at the leading edge (LE) of the blade. The stronger branch of the HV flowed over the pressure surface (PS), while the weaker branch flowed over the suction surface (SS). When the pressure-side horseshoe vortex (HV_ps_) interacted with the cross-flow in the passage channel, it created what is known as the PV. The PV induced the WV on the SS. Therefore, the CSV formed near the trailing edge (TE). A comparison of Case 1 with the ORI case revealed minimal changes in the passage vortex region, with similar results observed in Case 2. However, in Case 3, the placement of the fish scale array significantly reduced the size of the passage vortex and shifted it closer to the end-wall. Additionally, Case 3 showed a smaller wall vortex (WV) on the suction surface compared to the ORI case, a phenomenon not observed in Cases 1 and 2.

To investigate the suppression effects of secondary flow produced by the fish scale structure on the cascade blade, further analyses of the projected 2D streamlines and the velocity distribution were conducted to understand the mechanism of flow loss. Figure 13 shows a cross-section velocity distribution at a blade height of *z*/*s* = 4%, while Figure 14 illustrates the local flow structure and the projected 2D streamlines of the cascade. The results indicate that the climbing vortex generated by the fish scale structure produced excitation in the near flow field. In Case 1 and Case 2, the fish scale structure was not optimally positioned, leading to an increase in the region of low-energy fluid, similarly to the ORI case. By contrast, Case 3 exhibited the most substantial reduction in the region with low-energy fluid, demonstrating the lowest loss of the fish scale cascade. Under this condition, the fish scale structure generated climbing vortex rolls that extracted energy from the wall boundary layer and injected it into the passage vortex region. The kinetic energy near the passage vortex could enhance the energy transfer around it, thereby reducing the volume of the low-energy-fluid region and mitigating flow losses caused by the passage vortex.

To investigate the relationship between the low-energy-fluid region and the total pressure loss in the cascade, axial velocity density (AVD) and total pressure loss coefficient distribution were analyzed. Figure 15 shows the flow distribution of the total pressure loss coefficient in the cascade passage, with the low-energy fluid in the corner region identified by the iso-surface with an AVD of 5 kg/(s·m^2^). At 25% chord length, the total pressure loss in the cascade was concentrated near the end-wall due to the boundary layer. At 50% chord length, the PV and the WV expanded, leading to a flow loss region extending towards the blades and the emergence of a high-total-pressure-loss area closely associated with the vortex structures. Between 100% and 110% chord lengths, the concentrated shedding vortex (CSV) and trailing edge vortex (TEV) appeared in sequence, interacting at the trailing edge of the blade and intensifying the flow loss in the wake region. In this configuration, the low-energy-fluid regions near the fish scale structure were reduced, along with the areas of high total pressure loss. This result confirms the effectiveness of the fish scale structure in enhancing the kinetic energy of the fluid near the grooves and reducing total pressure loss.

In Case 1 and Case 2, no significant suppressive effect on the passage structure was discovered, leading to increased total pressure loss. However, in Case 3, the low-energy-fluid region was substantially reduced in the areas of the CSV and the PV, resulting in decreased spanwise penetration depth. The separation line (marked by a red dotted line) was closer to the end-wall in Case 3 than in the other cases, correlating with the lowest total pressure loss. Combined analysis of Figure 12 and Figure 15 revealed that low-energy-fluid regions were located precisely at the secondary flow positions, corresponding to high-total-pressure-loss areas. The findings demonstrate that the fish scale array greatly suppressed the PV and CV, reducing the total pressure loss in both the cascade and the groove.

Figure 16 shows the limiting streamlines on the suction surface; the solid orange line represents the separation line for the ORI case, the red dotted line indicates the separation line for the other cases, and *h_pd_* represents the spanwise penetration depth. The spanwise penetration depths in Case 1 and Case 2 were nearly identical to that of the ORI case, indicating no significant improvement. Conversely, Case 3 demonstrated a substantially reduced spanwise penetration depth compared to the ORI case, indicating a lower total pressure loss in this configuration. Overall, cases with diminished low-energy-fluid region and decreased total pressure loss also exhibited reduced spanwise penetration depth. This is consistent with previous findings by Wang et al. [30] and Benner et al. [14,15], highlighting the effectiveness of Case 3 in mitigating corner separation and enhancing cascade performance.

To analyze the distribution of the total pressure loss coefficient across all cases, eleven slices were created along the chord direction to obtain the total pressure loss distribution of the flow field, as illustrated in Figure 17. In Case 1, the fish scale structure failed to suppress the passage vortex structure and instead increased the total pressure loss by 0.4%. Similarly, in Case 2, the fish scale structure caused a slight increase in the total pressure loss by 0.3%. In contrast, Case 3 effectively suppressed both the passage vortex structure and the corner vortex structure, leading to a reduction in the low-energy-fluid region and achieving a 5.69% reduction in total pressure loss.

The computational domain was divided into four distinct areas to assess the distribution of total pressure loss across the cascades. These areas included the end-wall region, the passage region, the wake region, and the zone near the blade surface representing the blade profile. This division is illustrated in the three-dimensional (3D) diagram in Figure 18a. Figure 18b illustrates the total pressure loss in four regions for three cases. Each value represents the proportion of fish scale cases to ORI cases in the total pressure loss coefficient. In Case 1, the total pressure loss decreased in both the end-wall and wake regions but increased in the blade profile region. Case 2 showed reductions in the end-wall and passage regions, though losses in the blade profile region also increased. Notably, case 3 effectively reduced total pressure loss across all regions, particularly with reductions of 9.7% in the passage region and 3.35% in the end-wall region, indicating it as the most effective design configuration. This improvement is attributed to the placement of the bio-inspired fish scale structure ahead of the CV, which reduced the volume of the PV structure and the low-energy-fluid region near the end-wall, thereby enhancing overall cascade performance.

Figure 19 compares the limiting streamlines on the blade’s suction surface in Case 3, derived from numerical simulations and validated through experimental oil-film visualization. When the fish scale array was positioned ahead of the CV, the separation line observed in the numerical simulation closely matched the results from the oil-film visualization experiment. A comparison of Figure 7 and Figure 19 indicate that the fish scale array greatly suppressed the process of corner separation. Furthermore, the spanwise penetration depth in Case 3 was lower than in the ORI case, indicating a reduction in total pressure loss.

## 4. Conclusions

This study demonstrates the effectiveness of fish scale-inspired surface structures in reducing corner separation and improving aerodynamic performance within a compressor cascade. By applying the structures on the suction surface of the blade, specifically between 65% and 80% of the chord length, the climbing vortices generated by the geometry enhanced the kinetic energy of low-momentum fluid near the end wall, effectively suppressing both the passage vortex and the corner vortex. These modifications resulted in a shorter spanwise penetration depth and an overall reduction in total pressure loss by 5.69%. Region-specific improvements were particularly notable, with a 9.7% reduction in the passage region and a 3.5% reduction in the end-wall region, underscoring the structure’s effectiveness in addressing critical secondary flow losses.

Furthermore, the clear identification of separation lines along the suction surface provides useful insights for optimizing cascade geometry and improving aerodynamic performance. This biomimetic design demonstrates the potential of passive flow control using low-drag surface structures to enhance flow stability in compressor blade configurations. While the present study focuses on compressor cascades, the underlying design concept may be applicable to other fluid machinery where reducing secondary flow losses is important. Further investigations under varied operating conditions and with higher-fidelity turbulence models, such as large eddy simulation (LES), would help deepen the understanding of transition effects and complex vortex behaviors associated with such surface structures.

## Figures and Tables

**Figure 1 biomimetics-10-00473-f001:**
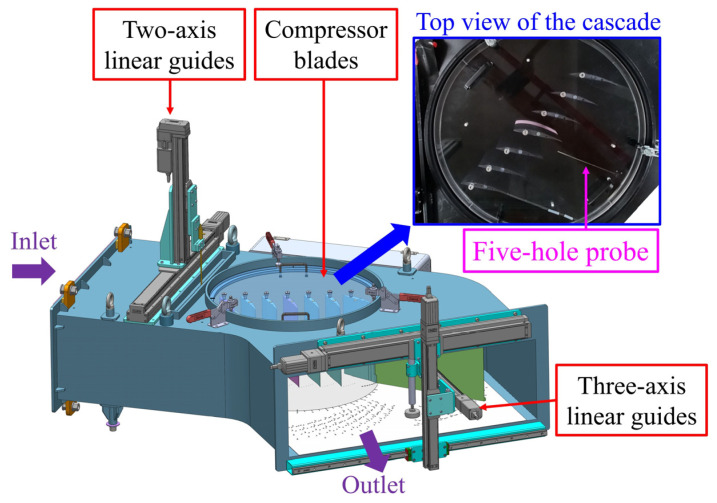
A schematic of the low-speed linear compressor cascade wind tunnel, the tailboard for maintaining periodic flow conditions, and a linear guide system for velocity measurement.

**Figure 2 biomimetics-10-00473-f002:**
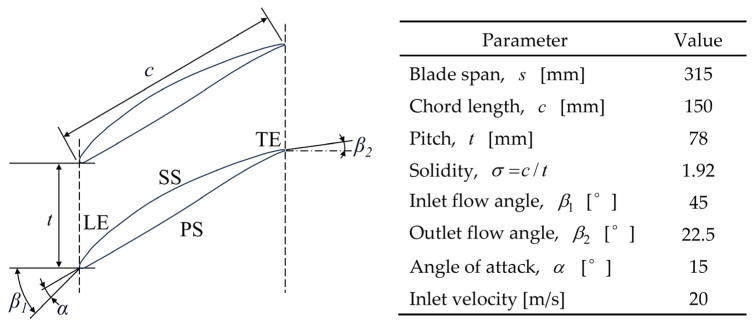
A schematic of the cascade blade geometry (**left**) and associated geometric parameters (**right**).

**Figure 3 biomimetics-10-00473-f003:**
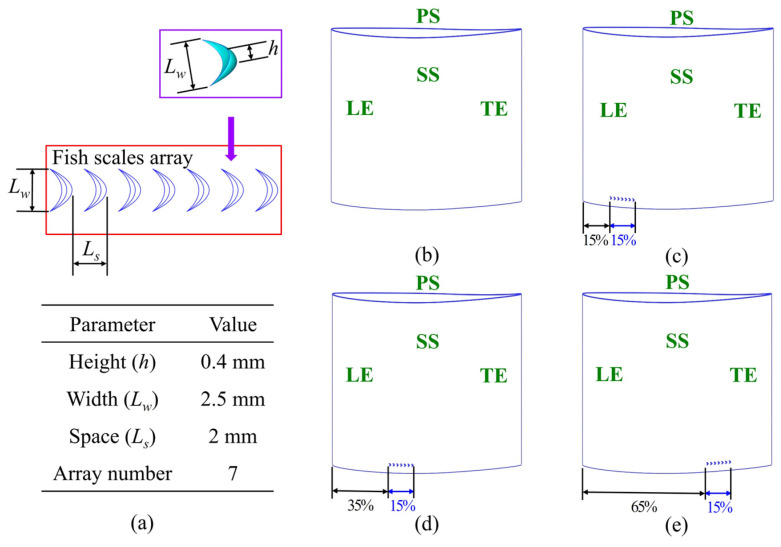
Configurations of cascades with a fish scale structure. (**a**) Geometric parameters of the fish scale. (**b**) An ORI case without fish scales. (**c**) Case 1: fish scales laid across 15–30% of the chord. (**d**) Case 2: fish scales laid across 35–50% of the chord. (**e**) Case 3: fish scales laid across 65–80% of the chord.

**Figure 4 biomimetics-10-00473-f004:**
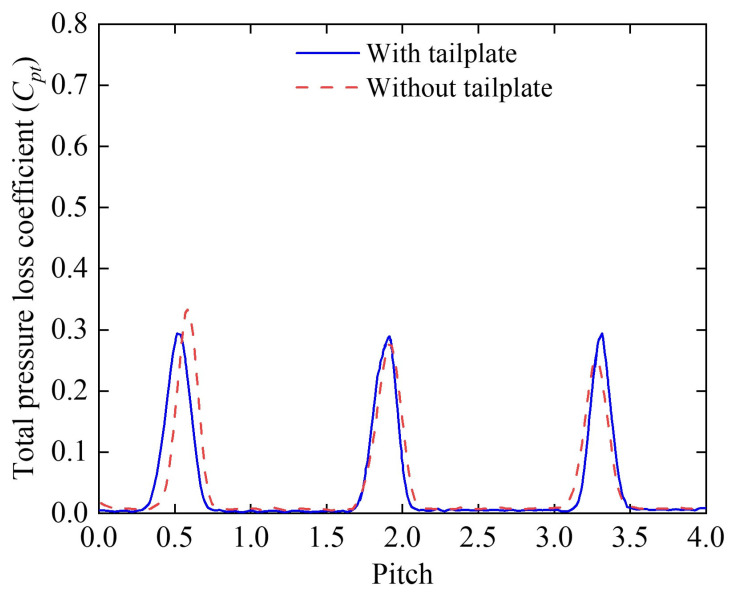
Total pressure loss distribution at the mid-span of the cascade outlet (ORI case) with *V* = 20 m/s and *α* = 15°, illustrating the flow periodicity along the grid pitch.

**Figure 5 biomimetics-10-00473-f005:**
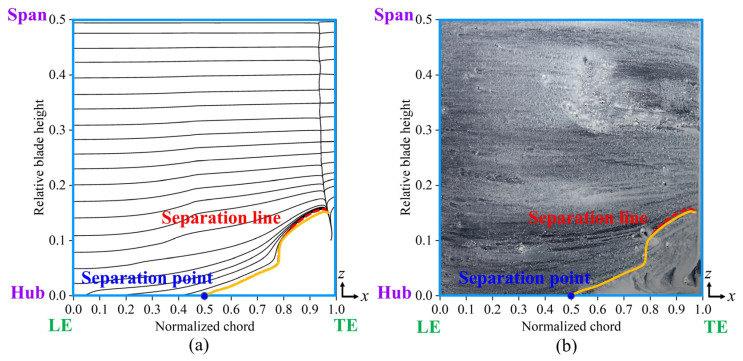
Limiting streamlines on the suction surface of the blade in the ORI case. (**a**) Numerical simulation results; (**b**) experimental oil-film visualization.

**Figure 6 biomimetics-10-00473-f006:**
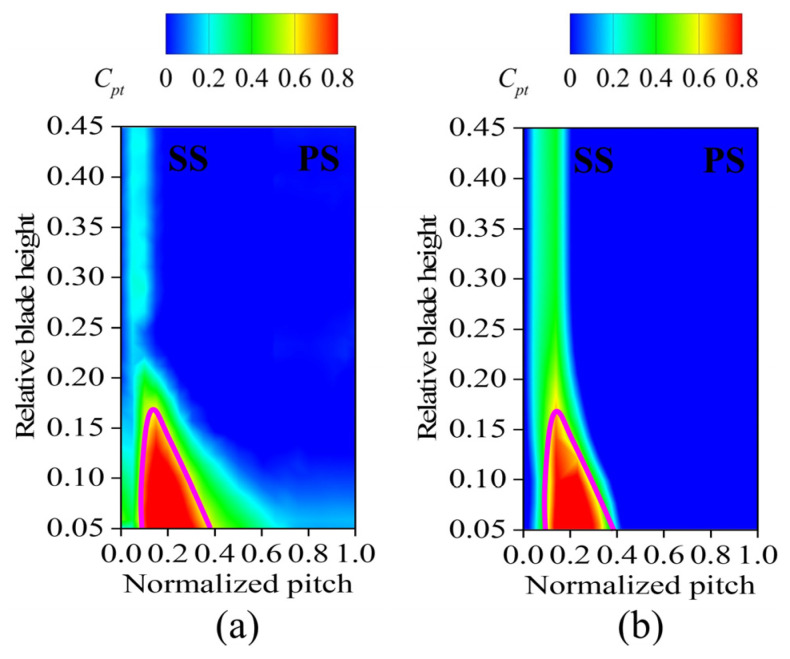
Total pressure loss coefficient distributions at *x*/*c* = 1.1 downstream of outflow in the ORI case. (**a**) Experimentally measured results; (**b**) numerically simulated results of the transient state.

**Figure 7 biomimetics-10-00473-f007:**
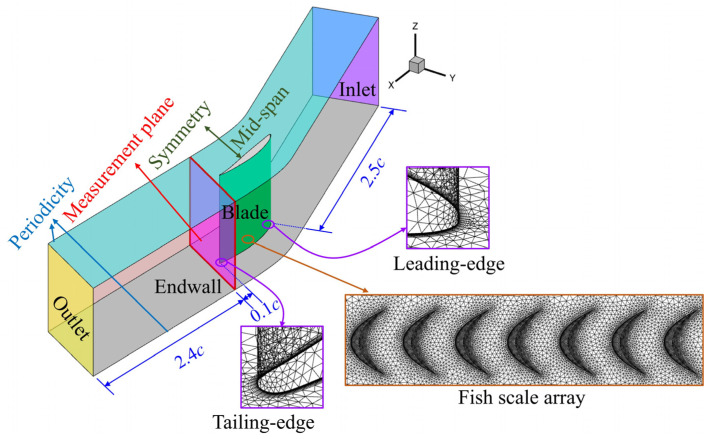
The computational grid and boundary conditions of the compressor cascade.

**Figure 8 biomimetics-10-00473-f008:**
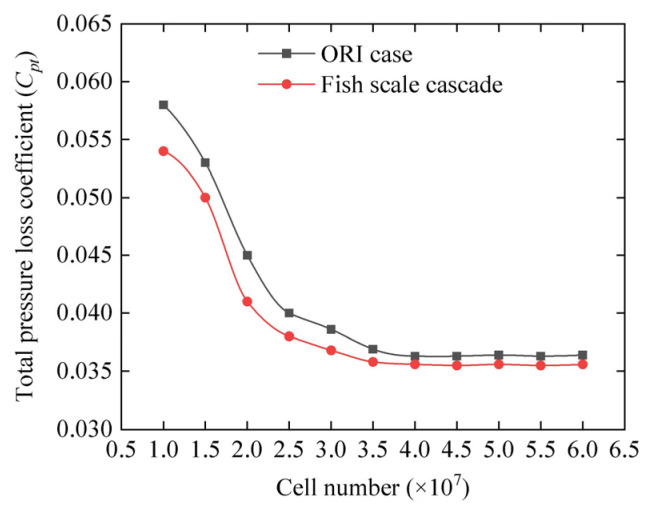
Grid independence test results showing the total pressure loss coefficient (*C_pt_*) at *x*/*c* = 1.1 downstream of the cascade.

**Figure 9 biomimetics-10-00473-f009:**
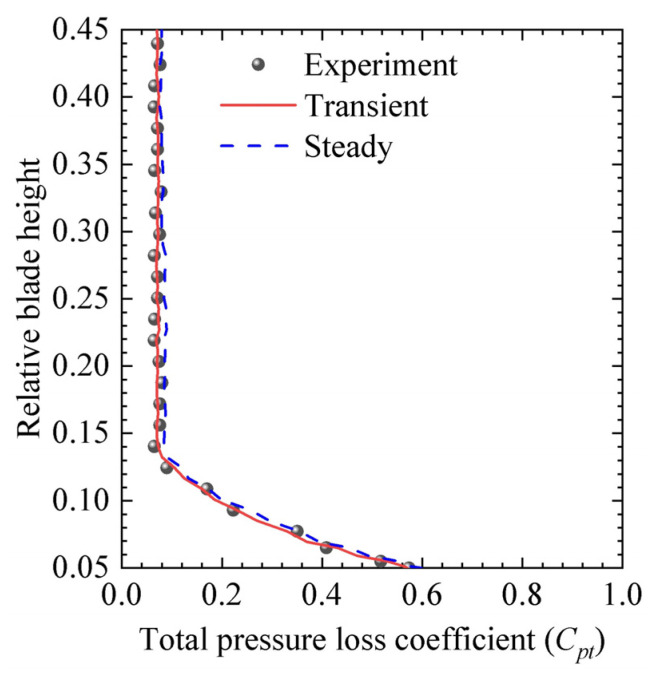
Comparisons of the spanwise mass-averaged total pressure loss coefficient (*C_pt_*) obtained from transient simulation at *t* = 0.07 s (time stepsize 10^−6^), steady-state simulation, and experimental measurements for the ORI case.

**Figure 10 biomimetics-10-00473-f010:**
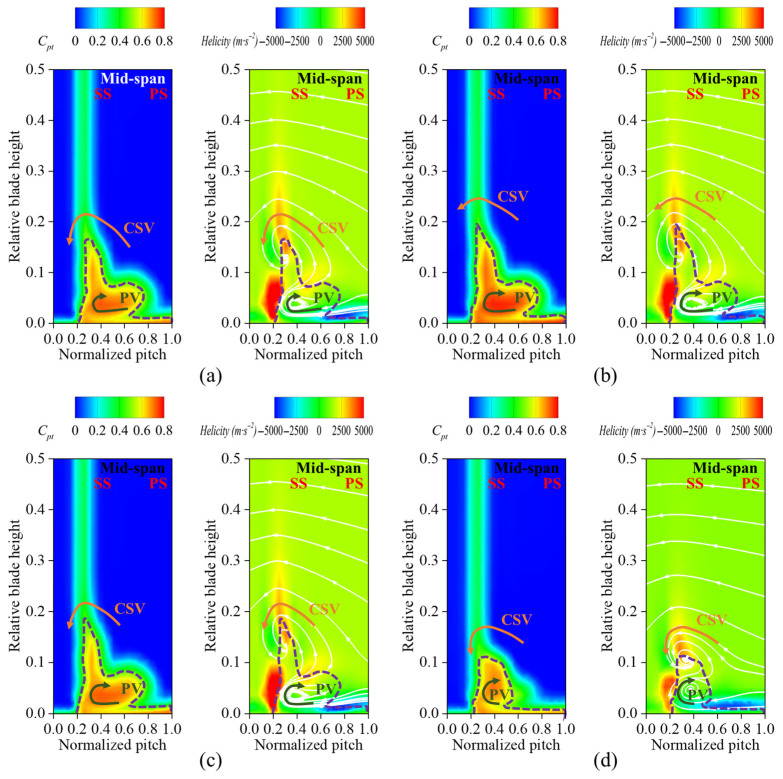
Contours of total pressure loss coefficient (*C_pt_*) and helicity (*H*) at the cascade outlet. (**a**) ORI case; (**b**) Case 1; (**c**) Case 2; (**d**) Case 3.

**Figure 11 biomimetics-10-00473-f011:**
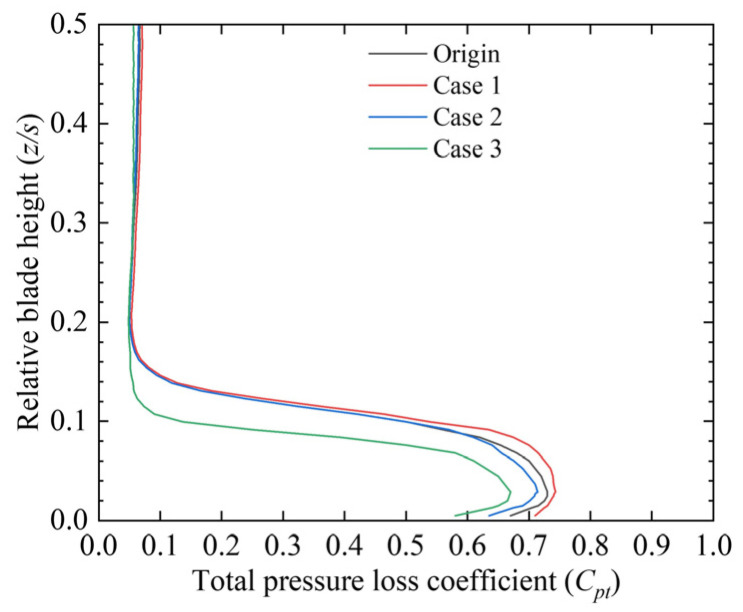
Spanwise distribution of the pitch-averaged total pressure loss coefficient (*C_pt_*) for the four cascade cases.

**Figure 12 biomimetics-10-00473-f012:**
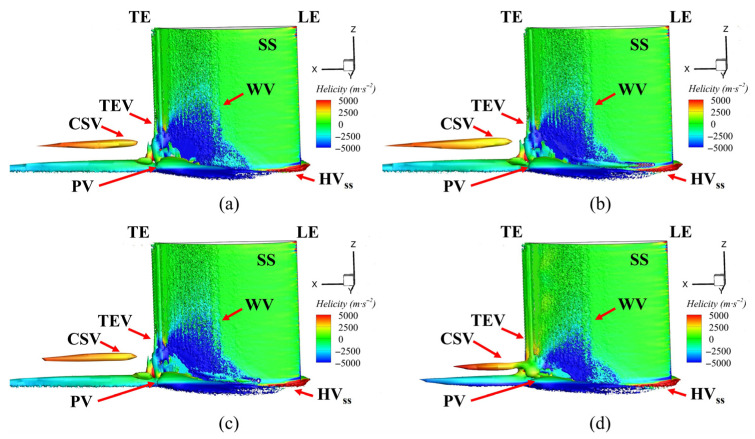
Three-dimensional vortex structure within the cascade. (**a**) ORI case; (**b**) Case 1; (**c**) Case 2; (**d**) Case 3.

**Figure 13 biomimetics-10-00473-f013:**
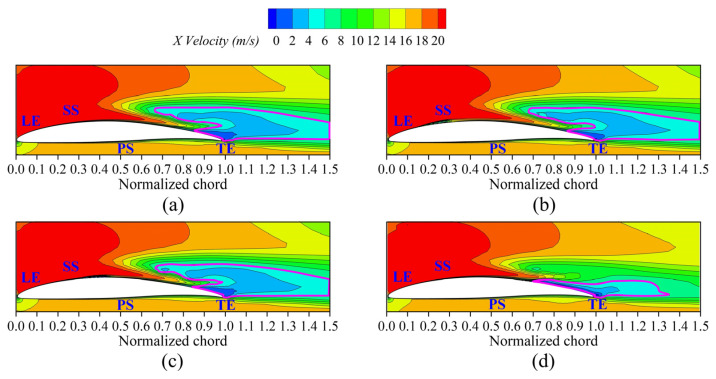
X-velocity contour at a blade height of *z*/*s* = 4%. (**a**) ORI case; (**b**) Case 1; (**c**) Case 2; (**d**) Case 3.

**Figure 14 biomimetics-10-00473-f014:**
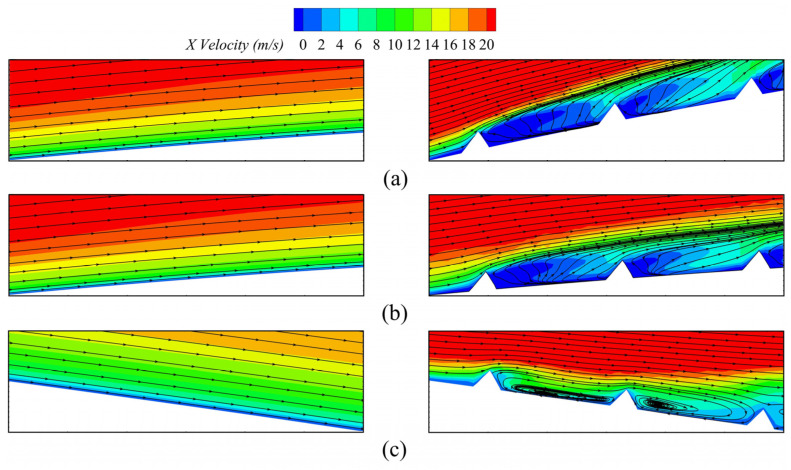
Local flow structure at a blade height of *z*/*s* = 4%. (**a**) ORI (**left**) and Case 1 (**right**); (**b**) ORI (**left**) and Case 2 (**right**); (**c**) ORI (**left**) and Case 3 (**right**).

**Figure 15 biomimetics-10-00473-f015:**
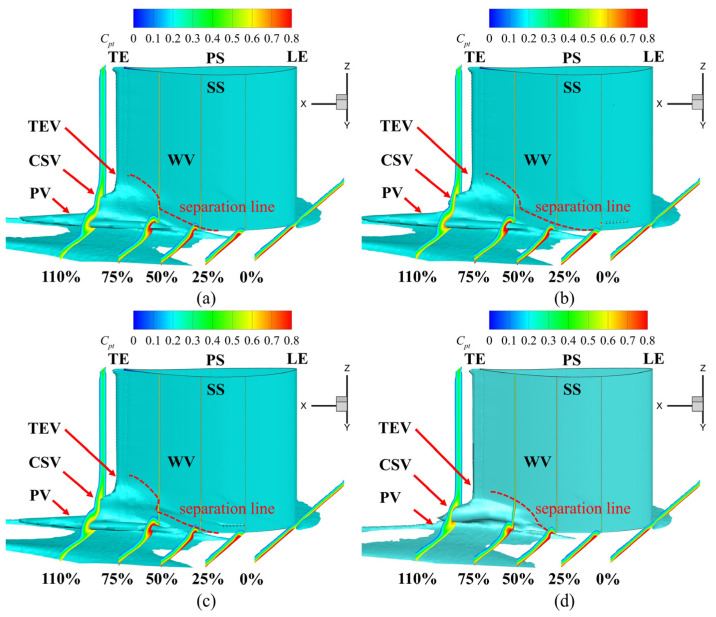
Total pressure loss coefficient along the flow direction with the cascade. (**a**) ORI case; (**b**) Case 1; (**c**) Case 2; (**d**) Case 3.

**Figure 16 biomimetics-10-00473-f016:**
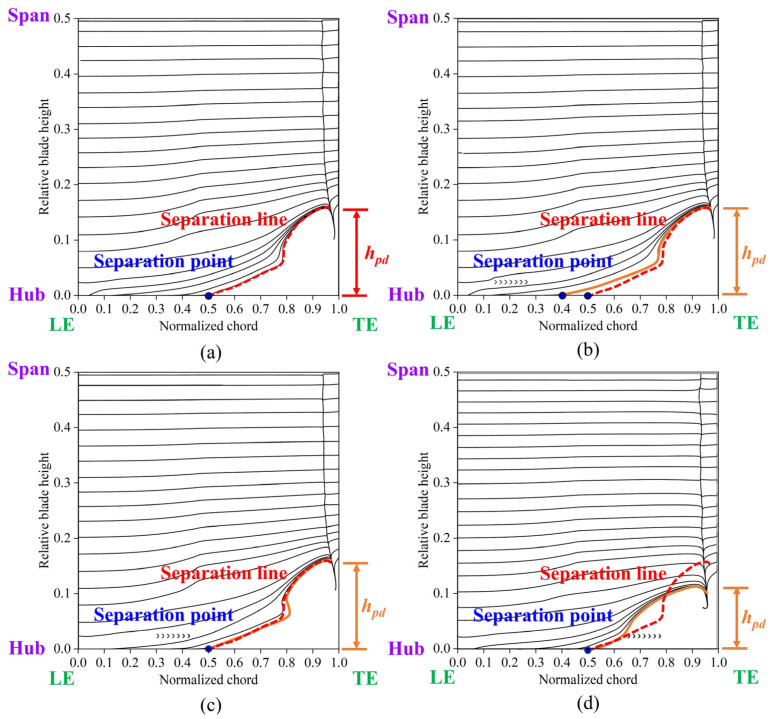
Numerical limiting streamlines on the suction surface in various cascade cases. (**a**) ORI case; (**b**) Case 1; (**c**) Case 2; (**d**) Case 3.

**Figure 17 biomimetics-10-00473-f017:**
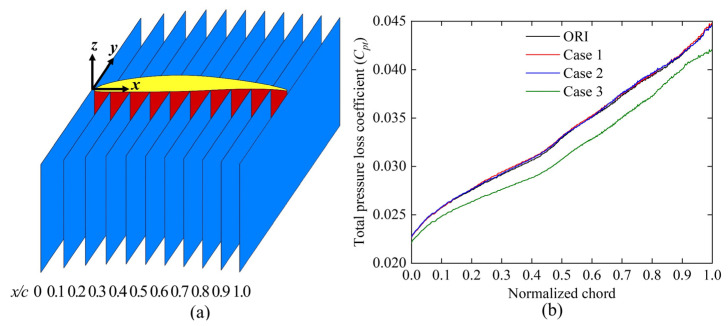
Distribution of total pressure loss coefficient in various cascade cases. (**a**) Slice surfaces along chord direction used to calculate total pressure loss coefficient; (**b**) total pressure loss coefficient along chord direction.

**Figure 18 biomimetics-10-00473-f018:**
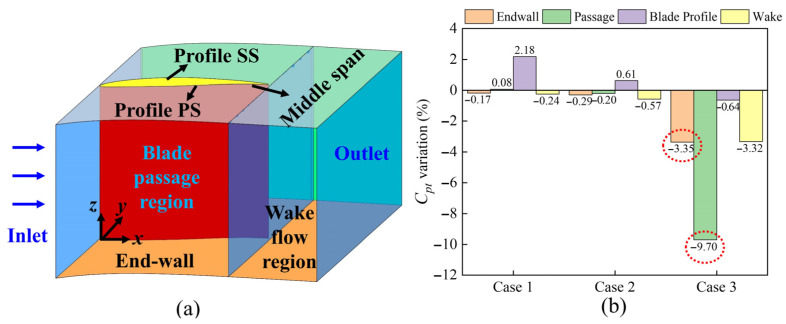
The proportion of four regions of loss sources. (**a**) Calculation regions; (**b**) the proportion of loss sources. The red circles highlight the regions with the most significant loss reductions in Case 3.

**Figure 19 biomimetics-10-00473-f019:**
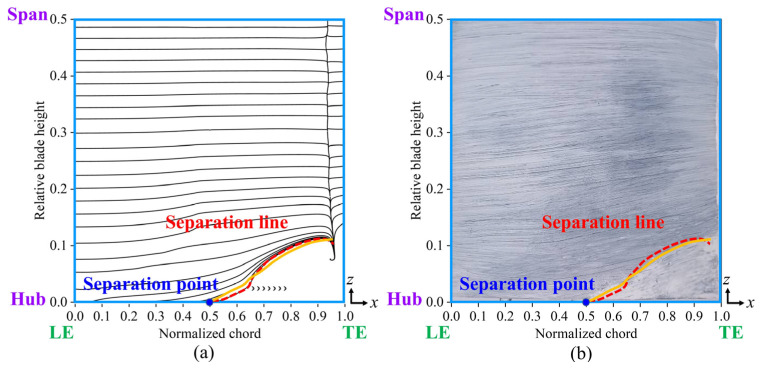
Limiting streamlines on the suction surface of the blade in Case 3. (**a**) Numerical limiting streamlines; (**b**) experimental oil-film visualization.

## Data Availability

The data supporting this study’s findings are available from the corresponding author upon reasonable request.

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
