# Peer review of "Fish Scale-Inspired Flow Control for Corner Vortex Suppression in Compressor Cascades"

_biomimetics, 2025, doi:10.3390/biomimetics10070473_

Round 1

Reviewer 1 Report

Comments and Suggestions for Authors

The paper is designed and written well. I have only minor comments here:

on page 2, line 79, it is much better to say that "we examined the literature on..."

page 5: Saying that RANS give "higher accuracy" is a bit odd as the community thins the other way. I suggest removing the phrase.

One general recommendation is to have a LES case to better capture the vortex structures and compare with RANS models.

Author Response

Please refer to the attached file for our point-by-point responses to the reviewer comments.

Reviewer 2 Report

Comments and Suggestions for Authors

The authors investigate both experimentally and numerically the effect of a biomimetic structure consisting of a fish-scale fin array on corner vortex suppression in a typical compressor cascade at design conditions. Their aim is to assess the effectiveness of such a passive measure on reducing total pressure loss. They use the location of the fin structure along the blade near the blade root as a parameter and mainly study its effect on the flow characteristics in terms of development of vortex  structures within the passage, since these are related to total pressure loss. Their work is very interesting, their approach is scientifically sound and the presentation of results is 'worthy of congratulations', since the decision of what to plot in order to visualize and assess the various vortices in the fully 3D field is not an easy task. The use of English language is very good and the overall manuscript is of rather high level.

In the light of the above, this reviewer suggests that the paper has to be published. However, some minor revision will be required, in order to make some  points clearer, according to the following remarks (P2/L79 means Page 2 / Line 79):

P2/L79: we examined --> they examined

P6/L191: cell counts --> number of cells

P7/L210-216: I think these lines should be transferred to P5/L172, where there is a discussion relevant to the selection of turbulence model (and not refer to it again in the validation section)

P7/L216: as our bionic cascade design --> for our bionic cascade investigation

Section 2.2.2: it should be explicitly stated that the validation refers to the case without the fish scale structures

P8/L226: Notably, a simulation time at --> Notably, for the transient simulation, a time instant at

Section 2.3: examine modifying the title to ‘Metrics for vortex visualization’

P9/L254-256: The last paragraph informs the reader that ‘Vortex identification will mainly be illustrated using the Q-Criterion technique, which highlights distinct iso-contours where the magnitude of vorticity surpasses that of the strain rates in the fluid.’ However, the plots in results section rather rely on plots of helicity and total pressure loss. Maybe the information on the use of Q-criterion has to be consistent with the presentation of results.

Throughout the text, the term ‘bionic’ is used. Do you think it is better to use the term ‘biomimetic’ or ‘bio-inspired’ instead of ‘bionic’??

Figure 9. Results from ‘steady’ and ‘transient’ simulations are referred in the text and the figure. However, it is not clear in the text if these are two different results or the transient calculation leads to a steady state result. It should be mentioned what simulations were made (transient, steady state, with what parameters and how the quantity plotted was calculated (steady value, time-averaged?).

P9/L261: clockwise) and projected --> clockwise), as well as projected

P9/L267-271: The text claims that results in Case 3 show to be affected by the presence of the fish scale structure. Although, this is clear by Figure 11, it is by no means obvious that the plots of Figure 10 support such a conclusion. The authors should rewrite this paragraph, if they reach such a conclusion based on the plots of Figure 10 (or at least help the reader to accept it and agree with their claim). For example, it seems that results by Case 1 (c) and Case 3 (d) are very similar.

Figure 10, legend: Clouds of --> Isosurfaces?

Author Response

(The authors gave the same response as above.)

Reviewer 3 Report

Comments and Suggestions for Authors

The paper is an excellent investigation of passive flow control. The application of fish scale-inspired passive flow control devices is analysed in detail through experimental and numerical means. The physical explanations are reasonable and correct.

It is recommended that the authors incorporate a section addressing turbulence and transition.

  1. The authors should add statements about transition in the case investigated. Were the computations performed without transition modelling? What's the condition of the boundary layer? Can these shape-like devices trigger the transition here?
  2. What were the typical Reynolds numbers of the "original biological fish scales" and the optimal fish scale structures found (Section 2.1.1)? Are they comparable?

Author Response

(The authors gave the same response as above.)
